# Translational control of ERK signaling through miRNA/4EHP-directed silencing

Seyed Mehdi Jafarnejad[1,2,†], Clément Chapat[1,2,†], Edna Matta-Camacho[1,2], Idit Anna Gelbart[3], Geoffrey G Hesketh[4], Meztli Arguello[1,2], Aitor Garzia[5], Sung-Hoon Kim[1,2], Jan Attig[6], Maayan Shapiro[1,2], Masahiro Morita[1,2,‡], Arkady Khoutorsky[7,8], Tommy Alain[9], Christos, G Gkogkas[10], Noam Stern-Ginossar[3], Thomas Tuschl[5], Anne-Claude Gingras[4,11], Thomas F Duchaine[1,2]*, Nahum Sonenberg[1,2]*

[1]Goodman Cancer Research Center, McGill University, Montréal, Canada; [2]Department of Biochemistry, McGill University, Montréal, Canada; [3]The Department of Molecular Genetics, Weizmann Institute of Science, Rehovot, Israel; [4]Centre for Systems Biology, Lunenfeld-Tanenbaum Research Institute, Sinai Health System, Toronto, Canada; [5]Laboratory for RNA Molecular Biology, Howard Hughes Medical Institute, The Rockefeller University, New York, United States; [6]The Francis Crick Institute, London, United Kingdom; [7]Department of Anesthesia, McGill University, Montréal, Canada; [8]Alan Edwards Centre for Research on Pain, McGill University, Montréal, Canada; [9]Children's Hospital of Eastern Ontario Research Institute, Department of Biochemistry, Microbiology and Immunology, University of Ottawa, Ottawa, Canada; [10]Patrick Wild Centre, Centre for Discovery Brain Sciences, University of Edinburgh, Edinburgh, United Kingdom; [11]Department of Molecular Genetics, University of Toronto, Toronto, Canada

*For correspondence:
thomas.duchaine@mcgill.ca (TFD);
nahum.sonenberg@mcgill.ca (NS)

[†]These authors contributed equally to this work

Present address: [‡]Department of Molecular Medicine and Barshop Institute for Longevity and Aging Studies, University of Texas Health Science Center at San Antonio, San Antonio, United States

**Abstract** MicroRNAs (miRNAs) exert a broad influence over gene expression by directing effector activities that impinge on translation and stability of mRNAs. We recently discovered that the cap-binding protein 4EHP is a key component of the mammalian miRNA-Induced Silencing Complex (miRISC), which mediates gene silencing. However, little is known about the mRNA repertoire that is controlled by the 4EHP/miRNA mechanism or its biological importance. Here, using ribosome profiling, we identify a subset of mRNAs that are translationally controlled by 4EHP. We show that the *Dusp6* mRNA, which encodes an ERK1/2 phosphatase, is translationally repressed by 4EHP and a specific miRNA, miR-145. This promotes ERK1/2 phosphorylation, resulting in augmented cell growth and reduced apoptosis. Our findings thus empirically define the integral role of translational repression in miRNA-induced gene silencing and reveal a critical function for this process in the control of the ERK signaling cascade in mammalian cells.
DOI: https://doi.org/10.7554/eLife.35034.001

## Introduction

mRNA translation commences with the binding of the eukaryotic initiation factor 4F (eIF4F) to the mRNA 5′ cap structure. eIF4F is a three-subunit complex composed of eIF4E, the m⁷GpppN (cap)-interacting factor; eIF4G, a scaffolding protein, and eIF4A, a DEAD-box RNA helicase (*Sonenberg and Hinnebusch, 2009*). eIF4G also interacts with eIF3, through which it recruits the pre-initiation complex, comprised of the 40S ribosomal subunit and associated factors, to the mRNA. Binding of the mRNA 5′ cap by the 4E Homologous Protein (4EHP, encoded by *Eif4e2*), in contrast to eIF4E, impairs translation initiation (*Cho et al., 2005*; *Morita et al., 2012*; *Rom et al.,*

1998). 4EHP shares 28% sequence identity with eIF4E (*Rom et al., 1998*) and is ubiquitously expressed, although it is 5–10 times less abundant than eIF4E in most cell types (*Joshi et al., 2004*). 4EHP binds the cap with 30- to 100-fold weaker affinity than eIF4E, but its affinity is increased by interactions with other proteins such as 4E-T or post-translational modification (*Chapat et al., 2017*; *Okumura et al., 2007*). 4EHP is involved in translational repression directed by miRNAs (*Chapat et al., 2017*; *Chen and Gao, 2017*). The miRNA-Induced Silencing Complex (miRISC) recruits the CCR4–NOT complex to effect mRNA translational repression and decay (*Jonas and Izaurralde, 2015*). CCR4–NOT in turn recruits DDX6, 4E-T (eIF4E-Transporter; a conserved 4EHP/eIF4E-binding protein) and 4EHP to suppress cap-dependent mRNA translation (*Chapat et al., 2017*; *Jonas and Izaurralde, 2015*; *Kamenska et al., 2014*; *Kamenska et al., 2016*; *Ozgur et al., 2015*). However, which cellular mRNAs are targeted by 4EHP remains unknown.

The Extracellular signal-Regulated Kinases (ERK1/2) are important effectors of the highly conserved Mitogen-Activated Protein Kinase (MAPK) signaling pathway (*Will et al., 2014*). ERK signaling is controlled by the RAS GTPase, which activates RAF, a serine/threonine kinase. RAF phosphorylates and activates the kinase MEK, which in turn phosphorylates and activates the effector serine/threonine kinases ERK1/2. Activated ERK signaling elicits multiple outcomes, including transcriptional programs that control cellular functions such as cell proliferation (*Aktas et al., 1997*; *Samatar and Poulikakos, 2014*), apoptosis (*Xia et al., 1995*) and mRNA translation (*Fukunaga and Hunter, 1997*).

Dual Specificity Phosphatase 6 (DUSP6), also called MAP Kinase Phosphatase-3 (MKP-3), is a highly specific phosphatase for ERK1/2 (*Caunt and Keyse, 2013*) and a key player in ERK signaling regulatory feedback loops (*Camps et al., 1998*; *Eblaghie et al., 2003*; *Kolch, 2005*; *Mendoza et al., 2011*). $Dusp6^{-/-}$ mice exhibit increased ERK1/2 phosphorylation at Thr202/Tyr204 residues (*Li et al., 2007*). DUSP6 expression is regulated transcriptionally (*Bermudez et al., 2011*; *Ekerot et al., 2008*; *Zhang et al., 2010*), and post-transcriptionally by miRNAs (*Banzhaf-Strathmann et al., 2014*; *Carson et al., 2017*; *Gu et al., 2015*) and RNA-binding proteins (*Bermudez et al., 2011*; *Galgano et al., 2008*; *Lee et al., 2006*). Altered expression or activity of DUSP6 impacts on ERK signaling in various diseases such as cancer and neurological disorders (*Banzhaf-Strathmann et al., 2014*; *Bermudez et al., 2008*; *Kawakami et al., 2003*; *Li et al., 2007*; *Molina et al., 2009*; *Pfuhlmann et al., 2017*; *Shojaee et al., 2015*).

Here, we employed ribosome profiling to identify a subset of mRNAs that are regulated by 4EHP. We discovered that *Dusp6* mRNA translation is repressed by a 4EHP/miRNA-dependent mechanism, which impacts on ERK1/2 phosphorylation, cell proliferation, and apoptosis. Our results underscore the biological importance of this translation repression mechanism, which is jointly orchestrated by miRNAs and 4EHP.

## Results

### Enrichment for miRNA-binding sites in 4EHP-regulated mRNAs

We recently discovered that 4EHP acts as a key component of the translational repression machinery, which is mobilized by miRNAs (*Chapat et al., 2017*). To identify mRNAs that are translationally controlled by 4EHP, we carried out ribosome profiling (*Ingolia et al., 2011*) in wild-type (WT) and 4EHP knockout (4EHP-KO) mouse embryonic fibroblasts (MEFs) (*Figure 1—figure Supplement 1A and B*). This assay measures the ribosome occupancy of each mRNA by deep sequencing of ribosome-protected mRNA fragments (ribosome footprints; RFPs) (*Ingolia et al., 2011*). We used the Babel tool (*Olshen et al., 2013*; *Stumpf et al., 2013*) to detect significant changes in translation efficiency (abundance of RFPs independently of changes in the levels of their corresponding mRNAs). Translation was up-regulated for 117 mRNAs (hereafter referred to as upregulated mRNAs) in 4EHP-KO in comparison to WT cells, while translation was down-regulated for 167 mRNAs (*Figure 1A* and *Supplementary file 1*). Whereas the translational up-regulation of the mRNAs can be explained by the activity of 4EHP as translational suppressor, translational downregulation may be the result of indirect adaptation effects following 4EHP loss.

We next analyzed the upregulated mRNAs for the presence of common sequence features in their UTRs or coding sequences. A significant positive correlation was observed between the length of the 3′ UTR and increased translation of the upregulated mRNAs in the 4EHP-KO cells (average of

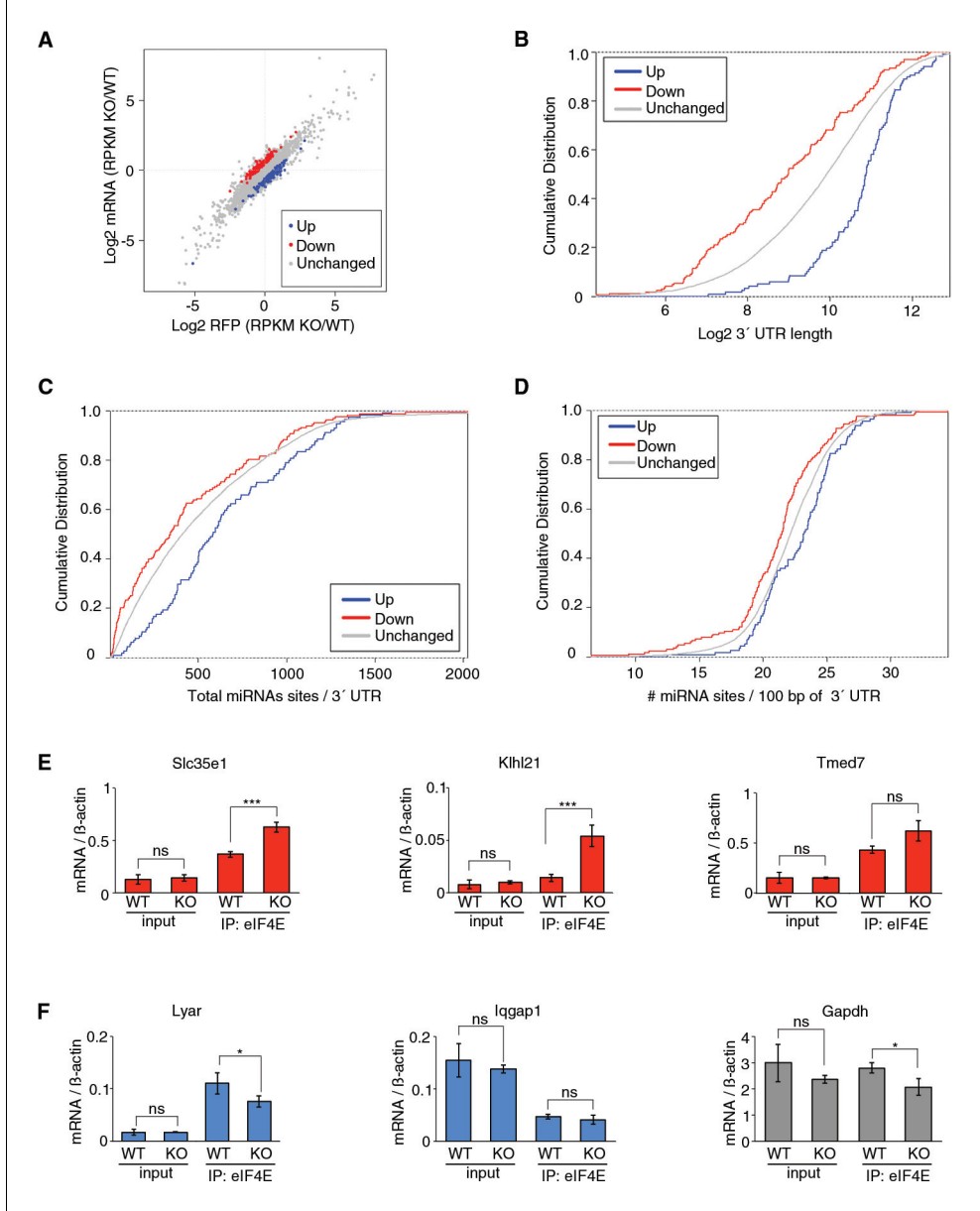

**Figure 1.** 4EHP controls translation of a subset of mRNAs. (**A**) The log2 ratio plot of abundance of ribosome footprints (RFP) and mRNAs in 4EHP-KO vs WT MEFs is shown. (**B**) Comparison of 3′ UTR length of mRNAs up- or down-regulated in 4EHP-KO MEFs. p-values: Up vs. Down: 2.26e-22, Up vs. Unchanged: 4.26e-17. (**C**) miRNA-binding sites in the 3′ UTR of mRNAs identified in (**A**). p-values: Up vs. Down: 0.000019, Up vs. Unchanged: 0.00040. (**D**) miRNA-binding site density (number of miRNA-binding sites per 100-nucleotide of 3′ UTR) in mRNA identified in (**A**). p-values: Up vs. Down: 0.000043, Up vs. Unchanged: 0.0063. (**E**) RNA-immunoprecipitation (RIP) analysis of the association of eIF4E with 4EHP targets in 4EHP-KO MEFs. eIF4E was immunoprecipitated using a monoclonal antibody against eIF4E from WT and 4EHP-KO MEFs. Levels of the indicated mRNAs (normalized to β-actin mRNA) in the inputs and eIF4E-bound mRNAs were analyzed by RT–qPCR. Data are mean ± SD (n = 3). The p-value was determined by two-tailed Student's t-test: (ns) non-significant, (*) p<0.05; (**) p<0.01; (***) p<0.001.

DOI: https://doi.org/10.7554/eLife.35034.002

The following figure supplement is available for figure 1:

**Figure supplement 1.** Analysis of 4EHP-sensitive mRNAs by ribosome profiling.
DOI: https://doi.org/10.7554/eLife.35034.003

2838.6, 2325.2, and 2016 nt for the up-regulated, unchanged and down-regulated mRNAs, respectively; p-value<2.2e-16; *Figure 1B*). We also found a less significant correlation (p=1.742e-05; *Figure 1—figure Supplement 1C*) between the length of the 5′ UTR and increased mRNA translation efficiency in the 4EHP-KO cells. This indicates that mRNAs with longer 3′ UTR are more likely to be translationally repressed by 4EHP.

mRNAs with long 3′ UTR generally contain more miRNA-binding sites (*Cheng et al., 2009*). We examined the number of miRNA-binding sites in the 3′ UTR of the up-regulated mRNAs (*Agarwal et al., 2015*). mRNAs which exhibit increased translation in 4EHP-KO cells, contained significantly more predicted miRNA-binding sites (642.8, 518.4, and 442.6 for the up-regulated, unchanged and down-regulated mRNAs, respectively; p-values: 0.0004, *Figure 1C*). We also calculated the density of miRNA-binding sites per 100-nucleotide of 3′ UTR and found 22.9, 22.1, and 21.1 for the up-regulated, unchanged and down-regulated mRNAs, respectively (p-values: 0.0063, *Figure 1D*), indicating a greater density of miRNA-binding sites in 3′ UTR of up-regulated mRNAs. These findings are in agreement with our previous report showing that 4EHP contributes to the translational silencing of miRNA targets by displacing eIF4E from the mRNA cap (*Chapat et al., 2017*). To verify that this mechanism affects the up-regulated mRNAs, we performed RNA immunoprecipitation (RIP) with an anti-eIF4E antibody in WT and 4EHP-KO MEFs. IP resulted in specific recovery of eIF4E (*Figure 1—figure Supplement 1D*). We examined the enrichment of the top three most translationally up-regulated mRNAs in 4EHP-KO cells (*Tmed7*, *Slc35e1* and *Klhl21*; *Supplementary file 1*) among the eIF4E-bound mRNAs (*Figure 1E*). *Slc35e1* and *Klhl21* but not *Tmed7* mRNAs were significantly enriched in eIF4E IP in 4EHP-KO cells in comparison with WT (*Figure 1E*). *Lyar* and *Iqgap1*, which were among the most significant translationally down-regulated mRNAs, were not enriched in eIF4E IP as a consequence of 4EHP loss (*Figure 1F*). These data show increased binding of eIF4E to the up-regulated mRNAs in 4EHP-KO cells, and indicate that 4EHP blocks the physical association of its target mRNAs with eIF4E.

## 4EHP-depletion impinges on cell viability and ERK1/2 phosphorylation

It was reported that while 4EHP expression is dispensable for growth in cell culture under physiological conditions, it is required under low oxygen conditions (*Uniacke et al., 2014*). However, at variance with these findings, we found that 4EHP-KO MEFs grew significantly slower than their WT counterparts (48 ± 3% less on day 6; p=0.002) under standard cell culture conditions (5% $CO_2$ and 20% $O_2$) (*Figure 2A*, *Figure 2—figure Supplement 1A and B*). Cell cycle analysis by FACS showed that the slow proliferation of 4EHP-KO cell populations is likely due to a decrease of the percentage of cells in S phase (30.3% and 21.4% for WT and KO cells, respectively; p=0.003), concomitant with an increase in the G0/G1 phase, compared with WT cells (50.2% and 57.7% for WT and KO cells, respectively; p=0.004, *Figure 2—figure Supplement 1C*). Consistently, depletion of 4EHP by shRNAs caused a dramatic reduction in proliferation of U251 (<90% at day 4; *Figure 2B*, *Figure 2—figure Supplement 1D*), and U-87 human glioblastoma cell lines (*Figure 2—figure Supplement 1E and F*). Notably, FACS analysis showed that unlike in MEFs, depletion of 4EHP in U251 cells increased the fraction of cells in sub-G1, which is associated with apoptosis (shCTR: 0.9%, sh4EHP#1: 15.5%, and sh4EHP#2: 11.4; *Figure 2C* and *Figure 2—figure Supplement 1G*). Accordingly, 4EHP depletion in U251 cells also induced the accumulation of cleaved-PARP (C-PARP), a marker of apoptosis (*Figure 2—figure Supplement 1D*).

The signaling pathways RAS/RAF/MEK/ERK and PI3K/mTOR control cell proliferation, growth and apoptosis, either in parallel or by converging on common downstream factors (*Cagnol and Chambard, 2010*; *Laplante and Sabatini, 2012*; *Mendoza et al., 2011*). We determined the phosphorylation levels of ERK1/2 and ribosomal protein S6 (RPS6) as respective markers of RAS/RAF/MEK/ERK and PI3K/mTOR activity by western blot (WB) analysis. While RPS6 phosphorylation remained unchanged, ERK1/2 phosphorylation (Thr202/Tyr204; pERK) was more than 80% reduced in 4EHP-KO MEFs in comparison with WT (*Figure 2D*). A similar result was obtained in U251 cells upon 4EHP-knockdown (*Figure 2—figure Supplement 1H*). However, phosphorylation of MEK, the immediate upstream kinase of ERK1/2, remained unchanged in 4EHP-depleted cells (*Figure 2D* and *Figure 2—figure Supplement 1H*). These results suggest that the expression or activity of a factor upstream of ERK1/2, which is independent of MEK, is deregulated in 4EHP-depleted cells.

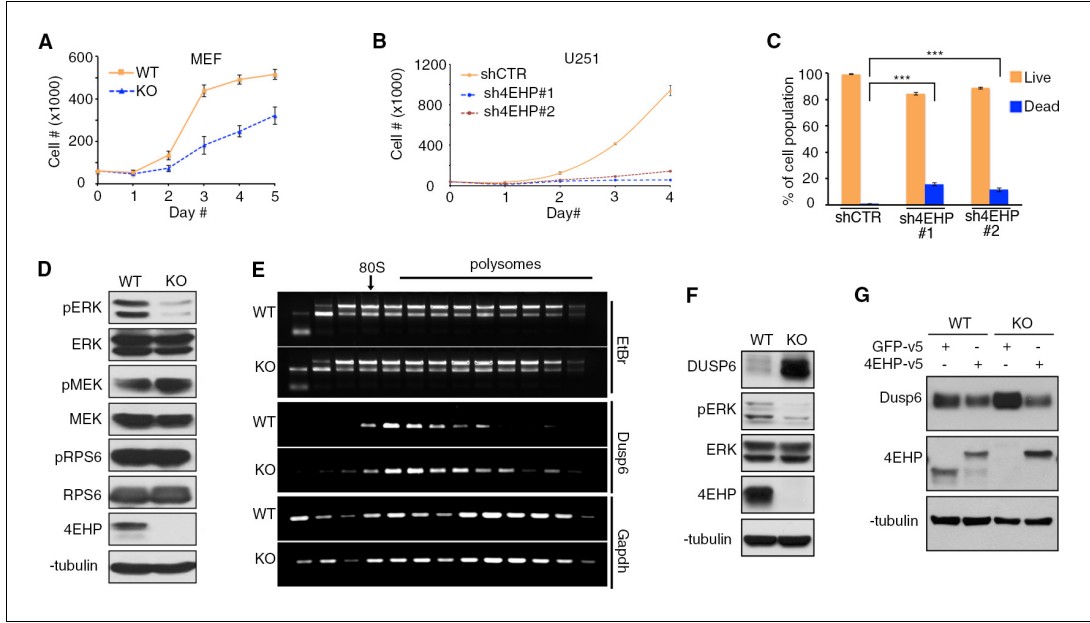

**Figure 2.** Depletion of 4EHP expression affects cell proliferation, survival, and ERK1/2 phosphorylation. (**A**) Cell proliferation assay. WT and 4EHP-KO MEFs were seeded in 6-well plates and trypsinized after the indicated time points and cell numbers determined using a hematocytometer. Data are mean ± SD (n = 3). (**B**) Cell proliferation assay. U251 cells with stable expression of shCTR (control), sh4EHP#1, and sh4EHP#2 were seeded in 6-well plates. Cells were trypsinized after the indicated time points and cell numbers determined using a hematocytometer. Data are mean ± SD (n = 3). (**C**) Quantitation of cell death by FACS assay; Sub-G population was considered as 'Dead' and G0/1, S and G2/M population was combined as 'Live'. Data are mean ± SD (n = 3). (**D**) WB for the indicated proteins in the WT and 4EHP-KO MEFs. (**E**) Polysome profiling/RT-PCR; RNA was extracted from each fraction (collected as described in *Figure 2—figure supplement 1J*), subjected to electrophoresis on agarose gel and visualized, using Ethidium Bromide (EtBr) staining. RT-PCR analyses of total RNA in each fraction was carried out with primers specific for *Dusp6* and *Gapdh* mRNAs. (**F**) WB on the indicated proteins in WT and 4EHP-KO MEFs. (**G**) WB for the indicated proteins in the WT and 4EHP-KO MEFs, expressing a v5-tagged GFP (GFP-v5) or v5-tagged 4EHP (4EHP-v5).

DOI: https://doi.org/10.7554/eLife.35034.004

The following figure supplement is available for figure 2:

**Figure supplement 1.** Cell proliferation and translational regulation of DUSP6 expression is affected by 4EHP depletion.

DOI: https://doi.org/10.7554/eLife.35034.005

## 4EHP represses *Dusp6* mRNA translation

We interrogated the 4EHP-KO MEF ribosome profiling data to identify candidate genes that could explain the strong impact of 4EHP on ERK1/2 phosphorylation. Interestingly, the mRNA encoding DUSP6, a potent and specific ERK1/2 phosphatase (*Caunt and Keyse, 2013*), was among the most translationally up-regulated transcripts in 4EHP-KO MEFs as compared to WT MEFs, with no significant change in its mRNA levels (*Supplementary file 1*). As expected, depletion of DUSP6 by shRNAs in U251 cells elicited ERK1/2 phosphorylation (*Figure 2—figure Supplement 1I*). To determine whether increased translation of *Dusp6* mRNA in 4EHP-KO MEFs is because of enhanced initiation, which is the rate limiting step in translation, we performed polysome profiling, which resolves mRNAs on a sucrose gradient according to the number of ribosomes with which they associate (*Figure 2—figure Supplement 1J*). While the distribution of the *Gapdh* mRNA along the sucrose gradient was similar in 4EHP-KO and WT cells, the *Dusp6* mRNA was shifted towards heavier fractions in the 4EHP-KO cells (*Figure 2E*), demonstrating augmented initiation. Consistent with greater translation efficiency, DUSP6 protein amount was markedly increased in 4EHP-KO MEF as compared to WT (*Figure 2F*). Up-regulation of DUSP6 protein level was also observed in U251 cells upon 4EHP knockdown in comparison with shCTR-treated cells (*Figure 2—figure Supplement 1K*). In contrast,

expression of DUSP7, another member of the DUSP phosphatase family, was not affected by 4EHP depletion (*Figure 2—figure Supplement 1L*), attesting to the specificity of 4EHP loss for mRNA translation. 4EHP depletion did not affect the abundance (*Figure 2—figure Supplement 1M*) or stability of *Dusp6* mRNA (*Figure 2—figure Supplement 1N*). Importantly, restoring 4EHP expression in 4EHP-KO MEFs significantly reduced DUSP6 protein levels (~3 fold repression; *Figure 2G*). Taken together, these data demonstrate that 4EHP controls expression of the ERK1/2 phosphatase DUSP6 at the level of mRNA translation initiation.

### *Dusp6* 3′ UTR confers translational sensitivity to 4EHP

To determine whether 4EHP regulates *Dusp6* translation by displacing eIF4E from the cap (*Chapat et al., 2017*; *Cho et al., 2005*), we examined the association of *Dusp6* mRNA with eIF4E in WT versus 4EHP-KO MEFs, using RIP. While *Dusp6* mRNA levels were not significantly different

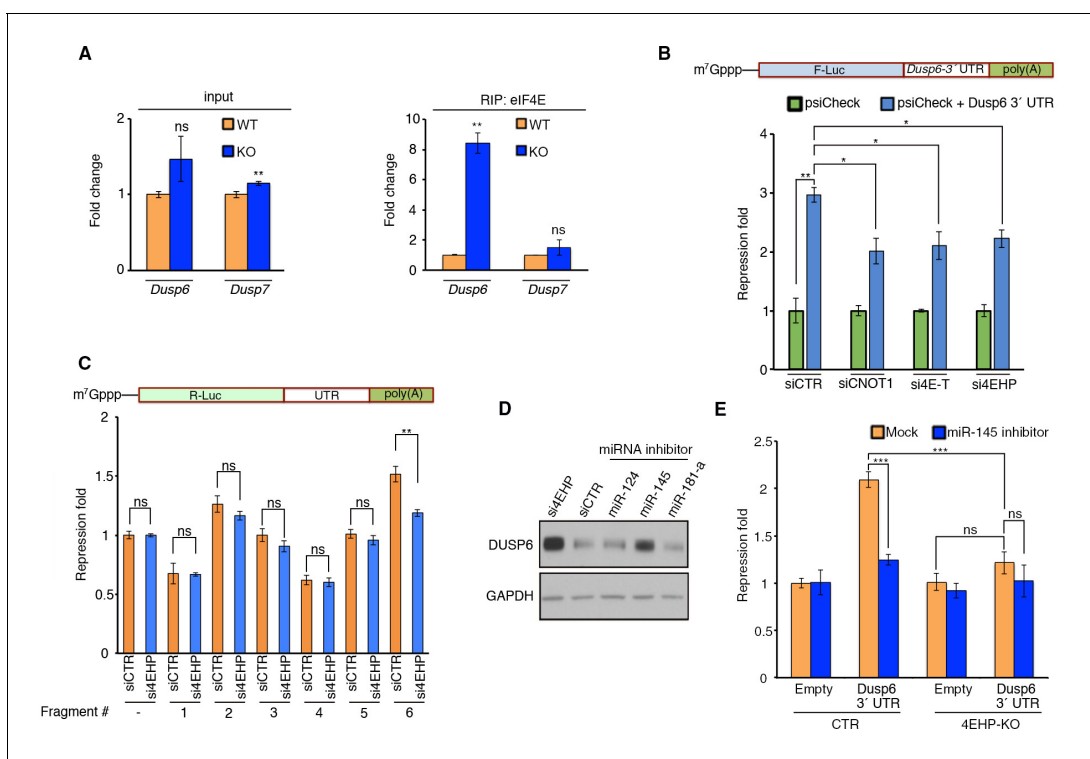

**Figure 3.** 4EHP enables miRNA-mediated silencing of *Dusp6* mRNA. (**A**) RIP analysis of the association of eIF4E with *Dusp6* mRNA in WT and 4EHP-KO MEFs. eIF4E was immunoprecipitated using a monoclonal antibody. Levels of the indicated mRNAs (normalized to *β-actin* mRNA) in the inputs and eIF4E-bound mRNAs were analyzed by RT–qPCR. Data are mean ± SD (n = 3). (**B**) *Top;* Schematic representation of the psiCHECK-FL-*Dusp6* 3′ UTR reporter. *Bottom;* CTR, CNOT1, 4E-T, or 4EHP-knockdown cells were co-transfected with psiCHECK-FL-*Dusp6* 3′ UTR reporter or the psiCHECK reporter (as control) in HEK293T cells. Luciferase activity was measured 24 hr after transfection. *Firefly* (*F-Luc*) values were normalized against *Renilla* (*R-Luc*) levels, and repression fold was calculated for the psiCHECK-FL-*Dusp6* 3′ UTR reporter relative to psiCHECK reporter level for each condition. Data are mean ± SD (n = 3). (**C**) The psiCHECK reporter (control) or psiCHECK-RL with truncated fragments of the *Dusp6* 3′ UTR were transfected into the HEK293T cells. Luciferase activity was measured 24 hr after transfection. *R-Luc* values were normalized against *F-Luc* levels, and repression fold was calculated for the psiCHECK-RL-*Dusp6* 3′ UTR reporter relative to psiCHECK reporter level for each condition. Data are mean ± SD (n = 3). (**D**) WB for the indicated proteins in U251 cells transfected with si4EHP or the indicated miRNA inhibitors. (**E**) The psiCHECK reporter (control) or psiCHECK-FL-*Dusp6* 3′ UTR were co-transfected along with the mock or miR-145 inhibitor in the control (CTR) or 4EHP-KO HEK293 cells. Luciferase activity was measured 24 hr after transfection. *F-Luc* values were normalized against *R-Luc* levels, and repression fold was calculated relative to the psiCHECK reporter/control inhibitor for each condition. Data are mean ± SD (n = 3). The *p*-values was determined by two-tailed Student's *t*-test: (ns) non-significant, (*) p<0.05; (**) p<0.01; (***) p<0.001.

DOI: https://doi.org/10.7554/eLife.35034.006

The following figure supplement is available for figure 3:

**Figure supplement 1.** Repression of DUSP6 expression by CCR4-NOT complex.
DOI: https://doi.org/10.7554/eLife.35034.007

between the WT and 4EHP-KO cells (*Figure 3A*; for corresponding WB analysis, see *Figure 1—figure Supplement 1D*), an 8-fold enrichment of *Dusp6* mRNA was detected in eIF4E IP from 4EHP-KO MEF lysates, as compared to WT (*Figure 3A*). As control, *Dusp7* mRNA was not enriched in eIF4E IP from 4EHP-KO MEFs lysates. These data lend further support to our model of displacement of eIF4E from the cap by 4EHP, and demonstrate that this mechanism causes translational repression of *Dusp6* mRNA.

3′ UTRs effect mRNA translation through trans-acting factors such as RNA-binding proteins (RBPs) and miRNAs (*Szostak and Gebauer, 2013*). DUSP6 expression is regulated by miRNAs including miR-145 (*Gu et al., 2015*), miR-181a (*Li et al., 2012*), and the RBP PUM2 (*Bermudez et al., 2011*), a homolog of *Drosophila* pumilio. We thus sought to study the role of the 3′ UTR of *Dusp6* mRNA in translational repression by 4EHP. To this end, 3′ rapid amplification of cDNA ends (3′ RACE) analysis was performed to amplify the 3′ UTR of *Dusp6* mRNA in U251 cells. A 1192-nucleotides segment was amplified (*Supplementary file 2*) and cloned into the psiCHECK-2 luciferase reporter vector. The resulting construct was transfected into HEK293T cells along with control siRNA (siCTR) or siRNA against 4EHP (si4EHP), or its partners CNOT1 (siCNOT1) and 4E-T (si4E-T). In the siCTR-transfected cells, the 3′ UTR of *Dusp6* mRNA caused a 3-fold repression in comparison with the backbone reporter alone (*Figure 3B*). However, knockdown of 4EHP or its partners CNOT1 and 4E-T significantly de-repressed the psiCHECK-*Dusp6*-3′ UTR reporter (38%, 49%, and 44% respectively as compared to siCTR; *Figure 3B*), thus supporting the role of CCR4-NOT/4E-T/4EHP pathway in *Dusp6* mRNA translational repression. Consistent with the latter results, knockdown of CNOT1 and CNOT9, two critical subunits of the CCR4-NOT complex, also led to an increase of DUSP6 protein amounts in U251 cells (1.4 and 2.2-folds, respectively; *Figure 3—figure Supplement 1A*).

We next mapped the repressive activity of 4EHP to elements of the 3′ UTR of *Dusp6* mRNA. To this end, we sub-cloned six ~200 nt fragments of the 3′ UTR into the psiCHECK-2 luciferase reporter (*Figure 3—figure Supplement 1B*). A segment harbouring both miR-145 and miR-181a binding sites exerted the strongest repression on the reporter (1.5 fold; p=0001, *Figure 3C*), which was alleviated upon 4EHP knockdown (*Figure 3C*). To identify which miRNA is responsible for repression of *Dusp6* mRNA, we used specific inhibitors to block miR-145, miR-181a, and miR-124 in U251 cells. While blocking miR-124 and miR-181a did not affect DUSP6 expression, a miR-145 inhibitor increased DUSP6 accumulation to a similar degree as knockdown of 4EHP (*Figure 3D*), without affecting the stability of the *Dusp6* mRNA (*Figure 3—figure Supplement 1C*). We further investigated the effect of miR-145 inhibitor on a luciferase reporter with the full-length *Dusp6* 3′ UTR. Unlike the control reporter, the expression of the reporter containing *Dusp6* 3′ UTR was significantly de-repressed in the presence of miR-145 inhibitor (1.25 fold repression compared with 2.09 for mock inhibitor; *Figure 3E*). Consistent with our observation that siRNA depletion of 4EHP in HEK293T cells de-repressed the *Dusp6* 3′ UTR reporter (*Figure 3B*), silencing of the same reporter was fully reversed in a 4EHP-KO HEK293 cells (*Figure 3E*). No de-repression by the miR-145 inhibitor was observed in 4EHP-KO HEK293 cells (*Figure 3E*). This confirms the requirement for 4EHP in miR-145-induced translational silencing of *Dusp6* mRNA. Taken together, these data demonstrate that the *Dusp6* mRNA translation is controlled by its 3′ UTR through the miRNA/CCR4-NOT/4E-T/4EHP pathway.

## De-repression of DUSP6 impedes ERK activity and proliferation in 4EHP-depleted cells

We next sought to determine the consequences of DUSP6 de-repression on ERK signaling and functions in 4EHP-KO MEFs. We used a selective small molecule inhibitor of DUSP6, 2-benzylidene-3-(cyclohexylamino)−1-Indanone hydrochloride (BCI) (*Molina et al., 2009*; *Shojaee et al., 2015*). Treatment of 4EHP-KO cells with BCI increased pERK1/2 to levels comparable with untreated WT cells within 30 min (*Figure 4A*). Similar results were obtained with U251 cells expressing an shRNA against 4EHP (*Figure 4—figure Supplement 1A*). These data confirm that reduced ERK1/2 phosphorylation in 4EHP-depleted cells is due to increased DUSP6 activity. Next, we examined the consequence of DUSP6 inhibition on proliferation of 4EHP-depleted cells by using shRNAs to knockdown DUSP6 in WT and 4EHP-KO cells (*Figure 4—figure Supplement 1B*). While DUSP6 knockdown did not have a detectable impact on WT cells proliferation, depletion of DUSP6 in 4EHP-KO cells markedly augmented their proliferation (42% increase for sh4EHP#1 [p=0.007] and 65%

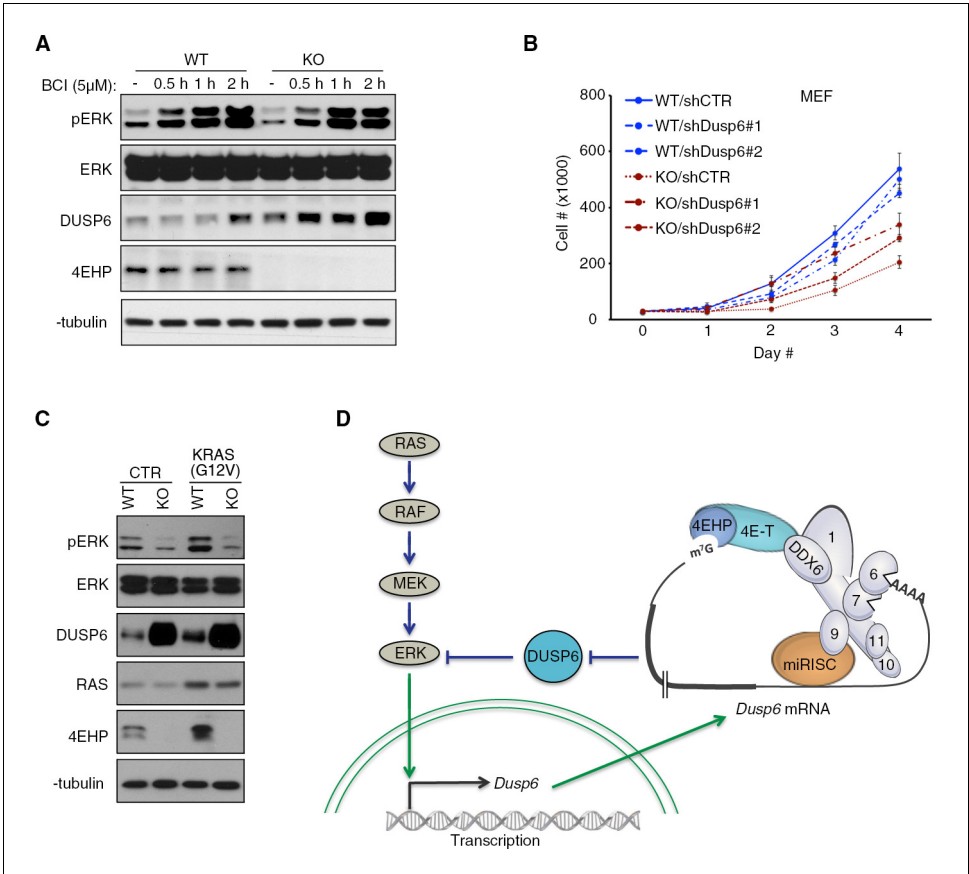

**Figure 4.** De-repression of DUSP6 in 4EHP-depleted cells impedes on ERK activity and cell proliferation. (A) Time course WB analyses of BCI-treated WT and 4EHP-KO MEFs. (B) Cell proliferation assay. WT and 4EHP-KO MEFs with stable expression of shCTR, shDusp6#1, and shDusp6#2 were seeded in 6-well plates. Cells were trypsinized after the indicated time points and cell numbers determined using a hematocytometer. Data are mean ± SD (n = 3). (C) WB for the indicated proteins in the WT and 4EHP-KO MEFs, with stable expression of a constitutively active mutant of KRAS (G12V). (D) Model of regulation of MAPK/ERK pathway activity by 4EHP through translational control of the *Dusp6* mRNA. Upon phosphorylation by MEK, ERK translocates to the nucleus and activates the *Dusp6* gene. The *Dusp6* transcript is then exported to the cytoplasm and translated. miRNAs control the translation of *Dusp6* mRNA via the CCR4-NOT/4E-T/4EHP complex and thus regulate the MAPK/ERK pathway activity.

DOI: https://doi.org/10.7554/eLife.35034.008

The following figure supplement is available for figure 4:

**Figure supplement 1.** DUSP6-mediated repression of ERK activity and cell proliferation in 4EHP-depleted cells.
DOI: https://doi.org/10.7554/eLife.35034.009

increase for sh4EHP#2 [p=0.004] on day 4; *Figure 4B*). This result demonstrates that the reduced proliferation of 4EHP-KO cells is at least partially due to de-repression of DUSP6.

Extracellular signals or mutations in *Ras* or *Raf*, which occur frequently in cancers, activate a phosphorylation cascade that results in phosphorylation and activation of ERK signaling (*Samatar and Poulikakos, 2014*). We examined whether 4EHP-depletion and the resulting increased DUSP6 expression could interfere with ERK1/2 phosphorylation in response to upstream activation of RAS. To this end, we expressed a constitutively active mutant KRAS (G12V) (*Prior et al., 2012*) and monitored ERK signaling by WB and proliferation assays. While ERK1/2 phosphorylation was increased by forced KRAS activity in WT MEFs, pERK levels remained unchanged in 4EHP-KO MEFs (*Figure 4C*). Consistent with these results, WT MEFs proliferation was slightly increased upon enforced KRAS activity, but remained unaffected in 4EHP-KO MEFs (*Figure 4—figure Supplement 1C*).

Taken together, the data demonstrate that 4EHP up-regulates ERK1/2 phosphorylation by effecting the miRNA-induced translational repression of *Dusp6* mRNA, and that depletion of 4EHP limits ERK activation by upstream signaling (*Figure 4D*, model).

## Discussion

We previously demonstrated that the cap-binding protein 4EHP acts as an effector of translational repression instigated by miRNAs. Here, we identify *Dusp6* mRNA as a functionally critical target of this silencing mechanism, which occurs in the absence of mRNA decay. Translational repression of *Dusp6* mRNA by the combined action of 4EHP and miR-145 down-regulates the MAPK/ERK signaling cascade and its output in cell proliferation and survival. The 4EHP/miRNA repression mechanism thus engenders important biological consequences in homeostasis and disease.

The relative contributions of translational repression and mRNA decay in miRNA-mediated silencing are in dispute. Several large-scale studies reported that mammalian miRNAs predominantly act by decreasing target mRNA levels (*Baek et al., 2008*; *Eichhorn et al., 2014*; *Guo et al., 2010*), while others showed that miRNAs affect the expression of target genes by translation inhibition (*Jin et al., 2017*; *Selbach et al., 2008*; *Yang et al., 2009*). It was convincingly demonstrated in *in vitro* and *in vivo* studies that translational repression precedes target mRNA decay (*Bazzini et al., 2012*; *Béthune et al., 2012*; *Djuranovic et al., 2012*; *Fabian et al., 2009*; *Mathonnet et al., 2007*). Because of their intricate nature, the exact contribution of either aspect of miRNA-mediated silencing in biological decisions has remained elusive. Our data demonstrate that 4EHP effects miRNA-mediated translational repression of *Dusp6* mRNA, but not mRNA stability. The relative contribution of translational repression and mRNA degradation to miRNA-mediated silencing may thus depend on the target mRNAs and on the cellular context. Expression of miRISC core and accessory components, post-translational modifications, translation efficiency, RNA structure within a 3′ UTR, or interactions with RNA-binding proteins (RBPs) may interfere or promote miRISC activities (*Cottrell et al., 2018*; *Cottrell et al., 2017*; *Kedde et al., 2010*; *Kundu et al., 2012*; *Long et al., 2007*). The RBPs PUM2 and TTP were implicated in the post-transcriptional repression of *Dusp6* mRNA, presumably in a CCR4-NOT-dependent mechanism (*Bermudez et al., 2011*; *Galgano et al., 2008*). Since the abundance of RBPs varies in tissues and under pathological conditions, it is conceivable that the potency and the nature of the miRNA-mediated silencing mechanism are modulated by such RBPs.

Our study underscores the importance of translational control in regulation of the ERK signaling pathway. Indirect up-regulation of ERK1/2 phosphorylation by 4EHP, via repression of *Dusp6* translation, explains the diminished cell proliferation in 4EHP-KO MEF cells and apoptosis observed in 4EHP-depleted U251 and U87 cells. A notable observation in our study is the impairment of the RAS/RAF/MEK/ERK pathway in 4EHP-depleted cells. Specifically, constitutively active RAS fails to increase ERK1/2 phosphorylation in 4EHP-KO MEFs. This can be explained by increased DUSP6 expression in 4EHP-KO cells, which effectively impairs phosphorylation of ERK1/2 downstream of RAS. Interestingly, over-expression of constitutively active RAS (*Park et al., 2014*), or BRAF (*Agrawal et al., 2014*), also induces DUSP6 expression constituting a negative feedback loop. The feedback loop restrains the activity of the RAS/RAF/MEK/ERK pathway upon induction by stimuli (e.g. growth factors). Thus, increasing DUSP6 expression by inhibiting 4EHP can potentially repress ERK pathway activation. While several pharmacological approaches have been described for targeting eIF4E (*Fischer, 2009*; *Graff et al., 2007*), to date no specific inhibitor of 4EHP has been discovered. The elucidation of the crystal structures of 4EHP in association with its binding partners (*Peter et al., 2017*; *Rosettani et al., 2007*) may prove useful for this purpose.

Our ribosome profiling data strongly suggest that translational repression through miRNA/4EHP impacts on many other mRNAs. An interesting miRNA to revisit in light of this mechanism is let-7, which suppresses tumorigenesis by directly silencing RAS expression (*Johnson et al., 2005*). We had previously shown that 4EHP contributes to the translational repression activity of a reporter mRNA by let-7 miRNA (*Chapat et al., 2017*), but let-7 miRNA can also clearly instigate mRNA deadenylation and decay. The relative contributions of translation repression and mRNA decay in the function of miRNA/mRNA pairs may be further revealed by systematically addressing their epistasis with 4EHP in the relevant cellular context.

## Materials and methods

### List of antibodies, siRNAs and shRNAs

The following antibodies were used: rabbit anti-eIF4E2 (4EHP) (Genetex, GTX103977), mouse anti-eIF4E (BD Biosciences, 610270), rabbit anti-eIF4ENIF1 (4E-T; abcam, ab55881), rabbit anti-DDX6 (Bethyl Laboratories, A300-460A), rabbit anti-CNOT1 (Proteintech, 14276–1-AP), mouse anti-α-Tubulin (Santa Cruz, sc-23948), mouse anti-β-actin (Sigma, A5441), mouse anti-Flag (Sigma, F3165), rabbit anti-HA (Sigma, H6908), mouse anti-V5 tag (Invitrogen, R960-25), rabbit anti-PARP (Cell Signaling Cat# 9532S), rabbit anti-DUSP6 (abcam Cat# ab76310), rabbit anti-DUSP7 (abcam Cat# ab100921),), rabbit anti-CNOT9 (RQCD1) (Proteintech Cat# 22503–1-AP), mouse GAPDH (Santa Cruz, sc-32233), rabbit anti-phospho-ERK1/2 (Thr202/Tyr204; Cell Signaling Cat#4370), mouse anti-MEK1/2 (Cell Signaling Cat# 4694S), rabbit anti-phospho-MEK1/2 (Ser217/221; Cell Signaling Cat# 9121S), rabbit anti-phospho-RPS6 (Ser240/244) (Cell Signaling Cat# 2215), and mouse anti-RPS6 (C-8).

The following siRNA and shRNAs were used: ON-TARGETplus Non-targeting Control Pool (Dharmacon, D-001810-10-05), 4EHP siRNA SMARTpool (Dharmacon, L-019870–01), eIF4ENIF1 (4E-T) siRNA SMARTpool (Dharmacon, L-013237–01), CNOT1 siRNA SMARTpool (Dharmacon, L-015369–01-0005), CNOT9 siRNA SMARTpool (Dharmacon, L-019972–00), Non-Targeting shRNA Controls (Sigma, SHC002), and EIF4E2 shRNA (Sigma, TRCN0000152006).

### Cell lines and culture conditions

MEFs, U251 (ATCC), U87 (ATCC), and HEK293T (Thermo Fisher Scientific, Waltham, MA) cells were maintained in DMEM supplemented with 10% foetal bovine serum and penicillin/streptomycin in a humidified atmosphere of 5% CO2 at 37°C. Control and 4EHP-knockout Flp-In T-REx 293 cells (HEK293, Thermo Fisher Scientific) were grown in high glucose DMEM (Thermo Fisher Scientific, 11965118) supplemented with 10% v/v FBS, 100 U/ml penicillin, 100 μg/ml streptomycin, 2 mM L-glutamine, 100 μg/ml zeocin and 15 μg/ml blasticidin. U251, U87, and HEK293T were tested for presence of mycoplasma contamination by LookOut Mycoplasma PCR Detection Kit (SIGMA, MP0035). Presence of mycoplasma in HEK293 cells was tested and dismissed by mRNA-Seq as previously described (*Garzia et al., 2017*).

### Inhibition of miRNA activity

The following miRNA inhibitors (Thermo Fisher Scientific, 4464084) were used: anti-miR-124 (MH10421), anti-miR-145 (MH11480), anti-miR-181 (MH10691) and mirVana negative control (4464076). 200,000 U251 cells were plated in a 6-well plate and transfected with a final concentration of 50 nM of each miRNA inhibitor for 72 hr using Lipofectamine 2000 (Invitrogen, Carlsbad, CA) according to the manufacturer's instructions.

### Lentivirus production

$8 \times 10^6$ HEK293 FT (Thermo Fisher Scientific, R70007) cells were cultured in a 10 cm dish for 24 hr in high glucose DMEM supplemented with 10% v/v FBS. Medium was replaced by OptiMEM (Thermo Fisher Scientific, 51985091) 30 min before transfection. Lentivirus particles were produced by transfecting the HEK293FT cells using Lipofectamine 2000 and 10 μg shRNA plasmid, 6.5 μg psPAX2 (Addgene, plasmid 12260) and 3.5 μg pMD2.G (Addgene, plasmid 12259) packaging plasmids. 5 hr post-transfection, the medium was replaced with fresh high glucose DMEM supplemented with 10% v/v FBS. Supernatant was collected at 48 hr post-transfection, replaced with fresh medium and collected after 24 hr. Viral particles were cleared by filtration (45 μm; Fisher Scientific, 09-720-005) and virus titer was measured by colony formation assay using 293FT cells. The multiplicity of infection (MOI) was adjusted to ~5. Virus solution was stored at −80°C without cryopreservative in 1 ml aliquots or used to infect the cells directly in the presence of 6 μg/ml polybrene (Sigma, H9268).

### CRISPR-Cas9 genome engineering for generating 4ehp knockout HEK293 cell line

CRISPR-Cas9-mediated genome editing of Flp-In T-REx HEK293 cells was performed as previously described (*Ran et al., 2013*). Two small guide RNAs (sgRNAs) cognate to the coding region of 4EHP gene: 5′-CAACAAGTTCGACGCGTGAG and 5′-TGAGCTCGTGGGACGGCCGG were designed. The

top and bottom strands of each designed sgRNA were annealed creating overhangs for cloning of the guide sequence oligos into pSpCas9(BB)−2A-GFP (Addgene, PX458, Plasmid #48138) by BbsI digestion. To generate gene knockout Flp-In T-REx HEK293 cells, we transfected 130.000 cells with the corresponding guide sequence containing pSpCas9(BB)−2A-GFP plasmid. 24 hr after transfection, GFP-positive single cells were sorted by FACS into 96-well plates and cultivated until colonies were obtained. Clonal cell lines were analyzed by WB for protein depletion as well as by PCR-genotyping. The following primers were used for the PCR-genotyping: sense primer1, 5´- GCCGCCC TGAGCTGGCGTCCC; anti-sense primer1, 5´- CGGCACAGCCACCCCTCCCCC; sense primer2, 5´- GCAGAATCTTTGGCACATTGCAGATAGTTGAGG; anti-sense primer2, 5´- GCCCTTCTGATCAACTC TACAATTCTCATATTTGTTGATACC. PCR products were cloned using the Zero Blunt PCR Cloning Kit (Thermo Fisher Scientific, K270040) and 10 clones sequenced per cell line.

## Real-Time RT-qPCR

1 µg of DNase I-treated total RNA, purified using the TRI-Reagent, was reverse-transcribed using 100 ng of random primers following the Superscript III (Invitrogen) protocol. Real time PCR was performed with SYBR Green master mix (iQ; Biorad) in a real-time PCR detection system (Mastercycler *Realplex*, Eppendorf). Mean values of triplicate measurements were calculated according to the $-\Delta\Delta$ Ct quantification method, and were normalized against the expression of the indicated mRNA. Specificity was confirmed by analyzing the melting curves of PCR products. RT-qPCR results were repeated at least three times in independent experiments and representative data sets are shown. Sequences of the used primers are listed in the *Supplementary file 3*.

## 3´ rapid amplification of cDNA ends (3´ RACE)

3' RACE was performed with the SMARTer RACE 5'/3' kit (Cat # 634858, Clontech, Mountain View, CA). 1 µg of total RNAs extracted from U251 cells was treated with DNase I (Fermentas) and cDNA was generated by the SMARTScribe Reverse Transcriptase (Clontech), according to the manufacturer's instructions. The resultant cDNA was used for PCR amplification using the human *DUSP6* gene-specific forward primers (GSPs) (*Supplementary file 2*) together with a common Universal Reverse Primer (UPM), provided by the manufacturer. PCR products were resolved by agarose gel electrophoresis and all visible bands were excised and digested by restriction enzymes followed by cloning into the PUC19 vector provided by the manufacturer and sequenced by Sanger sequencing.

## RNA immunoprecipitation (RIP)

RIP was performed as described previously (*Thoreen et al., 2012*) with minor modifications. WT and 4EHP-KO MEFs were seeded in $3 \times 15$ cm plates (at $10 \times 10^6$ cells per plate) and incubated overnight. Cells were lysed in lysis buffer A (50 mM HEPES-KOH (pH: 7.4), 2 mM EDTA, 10 mM pyrophosphate, 10 mM beta-glycerophosphate, 40 mM NaCl, 1% Trition X-100 and one tablet of EDTA-free protease inhibitors (Roche)) containing 40 U/ml SuperaseIn. Insoluble material was removed by centrifugation at 20,000xg for 5 min at 4°C. Protein concentration was measured by Bradford assay and 2 mg of lysate was pre-cleared by incubating with 50 µl of 50% protein G agarose fast flow beads (EMD Millipore, 16–266) for 2 hr at 4°C with gentle agitation. The cleared lysates were collected by centrifugation at 3000xg for 1 min at 4°C and collecting the supernatant. In parallel 2 µg of anti-eIF4E antibody was incubated with 50 µl of 50% protein G agarose fast flow beads for on an end-over-end rotator for 2 hr at 4°C. For IP, the pre-cleared lysates were incubated with the antibody + bead mixture, in 1 ml total volume on an end-over-end rotator for 2 hr at 4°C. The precipitated beads were then washed 3x with 1 ml buffer A, twice with buffer B (15 mM HEPES-KOH (pH 7.4), 7.5 mM MgCl2, 100 mM KCl, 2 mM DTT and 1.0% Triton X-100), and resuspended in 100 µl buffer B. 10 µl of the final mix was used for WB and the remaining was used for RNA extraction.

## Cycloheximide treatment and hypotonic cell lysis

Cells were pretreated with cycloheximide (Bioshop Canada Cat#CYC003) (100 µg/ml) for 5 min, and lysed in hypotonic buffer (5 mM Tris-HCl (pH 7.5), 2.5 mM MgCl2, 1.5 mM KCl, 1x protease inhibitor cocktail (EDTA-free), 100 µg/ml cycloheximide, 2 mM DTT, 200 U/ml RNaseIn, 0.5% (v/w) Triton X-100, and 0.5% (v/w) Sodium Deoxycholate), to isolate the polysomes.

## Collection of ribosome footprints (RFPs)

Ribosome profiling was performed as described (*Ingolia et al., 2012*), with minor modifications. Briefly, 500 µg of the ribonucleoproteins were subjected to ribosome footprinting by RNase I treatment at 4°C for 45 min with end-over-end rotation. Monosomes were pelleted by ultracentrifugation in a 34% sucrose cushion at 70,000xrpm for 3 hr and RNA fragments were extracted twice with acid phenol, once with chloroform, and precipitated with isopropanol in the presence of NaOAc and Gly-coBlue. Purified RNA was resolved on a denaturing 15% polyacrylamide-urea gel and the section corresponding to 28–32 nucleotides containing the RFPs was excised, eluted, and precipitated by isopropanol.

## Random RNA fragmentation and mRNA-Seq

100 µg of cytoplasmic RNA was used for mRNA-Seq analysis. Poly(A)+ mRNAs were purified using magnetic oligo-dT DynaBeads (Invitrogen) according to the manufacturer's instructions. Purified RNA was eluted from the beads and mixed with an equal volume of 2X alkaline fragmentation solution (2 mM EDTA, 10 mM Na2CO3, 90 mM NaHCO3, pH 9.2) and incubated for 20 min at 95°C. Fragmentation reactions were mixed with stop/precipitation solution (300 mM NaOAc pH 5.5 and GlycoBlue), followed by isopropanol precipitation. Fragmented mRNA was size-selected on a denaturing 10% polyacrylamide-urea gel and the area corresponding to 35–50 nucleotides was excised, eluted, and precipitated with isopropanol.

## Library preparation and sequencing

Fragmented mRNAs and RFPs were dephosphorylated using T4 polynucleotide kinase (New England Biolabs). Denatured fragments were resuspended in 10 mM Tris (pH 7) and quantified using the Bio-Analyzer Small RNA assay (Agilent). 10 pmol of RNA was ligated to the 3′-adaptor with T4 RNA ligase 1 (New England Biolabs) for 2 hr at 37°C. Reverse transcription was carried out using oNTI223 adapter (Illumina) and SuperScript III reverse transcriptase (Invitrogen) according to the manufacturer's instructions. Products were separated from the empty adaptor on a 10% polyacrylamide Tris/Borate/EDTA-urea (TBE-urea) gel and circularized by CircLigase (Epicentre). Ribosomal RNA amounts were reduced by subtractive hybridization using biotinylated rDNA complementary oligos (*Ingolia et al., 2012*). The mRNA and ribosome-footprint libraries were amplified by PCR (12 cycles) using indexed primers and quantified using the Agilent BioAnalyzer High-Sensitivity assay. DNA was then sequenced on the HiSeq 2000 platform with read length of 50 nucleotides (SR50) according to the manufacturer's instructions, with sequencing primer oNTI202 (5CGACAGGTTCAGAGTTC TACAGTCCGACGATC).

## Analysis of ribosome profiling data

Prior to alignment, linker and polyA sequences were removed from the 3′ ends of reads. Bowtie v0.12.7 (allowing up to two mismatches) was used to perform the alignments. First, reads that aligned to rRNA sequences were discarded. All remaining reads were aligned to the mouse genome (mm10). Finally, still-unaligned reads were aligned to the mouse known canonical transcriptome that includes splice junctions. Reads with unique alignments were used to compute the total number of reads at each position. Footprints and mRNA densities were calculated in units of reads per kilobase per million (RPKM) to normalize for gene length and total reads per sequencing run. The expression patterns were examined for genes that had more than 150 uniquely aligned reads of mRNA and footprints in one of the samples. The Babel computational framework was used to quantitatively evaluate if there are genes that are differently translated in KO cells. The 5′ and 3′ UTRs were obtained from the UCSC Genome Browser. For translationally induced or repressed genes the length of 5′ and 3′ UTRs were calculated and compared using Welch Two Sample t-test. Predicted miRNA sites were retrieved from TargetScanMouse. Both conserved and non-conserved sites were taken into account. The number of miRNA sites per 100 bp of 3′ UTR was calculated using the 3′ UTR lengths published on TargetScanMouse. The GEO accession numbers for the sequencing data reported in this paper is GSE107826.

## RNA stability assay

300,000 cells were plated in 6-well plates and 5 µg/ml actinomycin D (Sigma) was added to the culture medium at the indicated times. RNA was isolated by using Tri Reagent (Sigma-Aldrich, St. Louis, Missouri), according to the manufacturer's protocol and the stability of the indicated transcript was measured by RT-qPCR with the primers indicated in *Supplementary file 3*.

## Preparation of reporter constructs

To generate luciferase reporter plasmids, a modified version of psiCHECK-2 (Promega, Madison, WI) containing the Gateway cassette C.1 (Invitrogen) at the 3′ end of the firefly luciferase (*F-Luc*) gene was used as described before (*Suffert et al., 2011*). The 3′ UTR sequence of *Dusp6* mRNA inserted in the PUC19 vector was obtained from the U251 cells by 3′ RACE assay. The *att*B-Dusp6 fragment was obtained by PCR with the primers indicated in *Supplementary file 3*, cloned into pDONR/Zeo (Invitrogen) and recombined in the modified psiCHECK-2 vector by Gateway cloning. The fragments of the 3′ UTR of *Dusp6* were obtained by PCR from the psiCHECK-Dusp6 3′ UTR vector and inserted as XhoI-NotI fragments into the psiCHECK-2 vector at the 3′-end of the *Renilla* luciferase gene (*R-Luc*). Sequences of the used primers are listed in the *Supplementary file 3*.

## Luciferase reporter assay

HEK293T and U251 cells (150,000 cells/well) were co-transfected in a 24-well plate with 10 ng psiCHECK-Dusp6 3′ UTR. For 4EHP knockdown, $4 \times 10^6$ cells were plated in a 10 cm culture dish and transfected with a final concentration of 25 nM of siRNA duplexes using Lipofectamine 2000 according to the manufacturer's instructions. After 24 hr, cells were plated in a 24-well plate and transfected a second time with the psiCHECK vectors as described above. Cells were lysed 24 hr after transfection. Luciferase activities were measured with the Dual-Luciferase Reporter Assay System (Promega) in a GloMax 20/20 luminometer (Promega). For experiments with miRNA inhibitors, HEK293 cells were co-transfected in a 24-well plate with 10 ng psiCHECK-Dusp6 3′ UTR and miRNA inhibitors were added to the transfection mixture at a final concentration of 50 nM.

## Acknowledgements

We thank Owen Cheng for technical assistance; Joshua Dunn, and Nadeem Siddiqui for discussions and Chris Rouya for reagents. The work was supported by a Canadian Institute of Health Research (CIHR) Foundation grant FDN-148423 (to NS), FDN-143301 (to A-CG), and MOP-123352 (to TFD.), the Fonds de la Rercherche en Santé du Québec (FRSQ), Chercheur-Boursier Senior salary award (TFD), and Natural Sciences and Engineering Research Council of Canada (NSERC; RGPIN-2014–06434 to A-CG). SMJ is a recipient of McLaughlin and CIHR Postdoctoral fellowships. CC is supported by FRQS and Fondation pour la Recherche Médicale (FRM) postdoctoral fellowships. GGH was supported by a Parkinson Canada Basic Research Fellowship. A-CG is the Canada Research Chair in Functional Proteomics.

## Additional information

#### Competing interests

Nahum Sonenberg: Reviewing editor, *eLife*. The other authors declare that no competing interests exist.

#### Funding

| Funder | Grant reference number | Author |
|---|---|---|
| Canadian Institutes of Health Research | FDN-148423 | Nahum Sonenberg |
| Fonds de la Recherche en Sante du Quebec | | Thomas F Duchaine |

| Natural Sciences and Engineering Research Council of Canada | RGPIN-2014-06434 | Anne-Claude Gingras |
| Canadian Institutes of Health Research | FDN-143301 | Anne-Claude Gingras |
| Canadian Institutes of Health Research | MOP-123352 | Thomas F Duchaine |

The funders had no role in study design, data collection and interpretation, or the decision to submit the work for publication.

### Author contributions

Seyed Mehdi Jafarnejad, Conceptualization, Resources, Data curation, Formal analysis, Validation, Investigation, Visualization, Methodology, Writing—original draft, Project administration, Writing—review and editing; Clément Chapat, Resources, Data curation, Formal analysis, Validation, Investigation, Visualization, Methodology, Writing—original draft, Writing—review and editing; Edna Matta-Camacho, Investigation, Methodology, Writing—review and editing; Idit Anna Gelbart, Data curation, Formal analysis, Visualization, Methodology, Writing—review and editing; Geoffrey G Hesketh, Aitor Garzia, Resources, Data curation, Writing—review and editing; Meztli Arguello, Data curation, Methodology, Writing—review and editing; Sung-Hoon Kim, Maayan Shapiro, Resources, Investigation, Writing—review and editing; Jan Attig, Resources, Data curation, Visualization, Writing—review and editing; Masahiro Morita, Anne-Claude Gingras, Resources, Writing—review and editing; Arkady Khoutorsky, Christos, G Gkogkas, Resources, Methodology, Writing—review and editing; Tommy Alain, Resources, Investigation, Visualization, Writing—review and editing; Noam Stern-Ginossar, Resources, Formal analysis, Visualization, Writing—review and editing; Thomas Tuschl, Resources, Supervision, Writing—review and editing; Thomas F Duchaine, Supervision, Funding acquisition, Project administration, Writing—review and editing; Nahum Sonenberg, Conceptualization, Supervision, Funding acquisition, Writing—review and editing

### Author ORCIDs

Seyed Mehdi Jafarnejad (iD) http://orcid.org/0000-0002-5129-7081
Clément Chapat (iD) http://orcid.org/0000-0002-5806-7959
Jan Attig (iD) http://orcid.org/0000-0002-2159-2880
Christos, G Gkogkas (iD) https://orcid.org/0000-0001-6281-3419
Noam Stern-Ginossar (iD) https://orcid.org/0000-0003-3583-5932
Nahum Sonenberg (iD) http://orcid.org/0000-0002-4707-8759

### Decision letter and Author response

Decision letter https://doi.org/10.7554/eLife.35034.017

# Additional files

### Supplementary files

• Supplementary file 1. mRNAs differentially translated in 4EHP-KO vs WT MEFs identified by the ribosome profiling assay.
DOI: https://doi.org/10.7554/eLife.35034.010

• Supplementary file 2. Dusp6 3′ UTR isolated from U251 human glioblastoma cell line. Highlighted sequence represent the translation stop codon.
DOI: https://doi.org/10.7554/eLife.35034.011

• Supplementary file 3. List of primers used in this study.
DOI: https://doi.org/10.7554/eLife.35034.012

• Transparent reporting form
DOI: https://doi.org/10.7554/eLife.35034.013

### Major datasets

The following dataset was generated:

| Author(s) | Year | Dataset title | Dataset URL | Database, license, and accessibility information |
|---|---|---|---|---|
| Sonenberg N | 2017 | Translational control of ERK signalling pathway by the mRNA cap-binding protein 4EHP | https://www.ncbi.nlm.nih.gov/geo/query/acc.cgi?acc=GSE107826 | Publicly available at the NCBI Gene Expression Omnibus (accession no: GSE107826). |

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
