## [Decision Letter]

[Editors’ note: minor issues and corrections have not been included, so there is not an accompanying Author response.]

Congratulations, we are pleased to inform you that your article, "Translational control of ERK signaling through miRNA/4EHP-directed silencing", has been accepted for publication in *eLife*.

All three reviewers recognise the significance of the discovery of a physiologically-important target for miR-CCR4/NOT-4ET-4EHP axis and the rigour with which the mechanistic aspects have been researched. We recommend acceptance of the manuscript in its current form, and leave it to the authors if they do or do not wish to take stock of the various comments and revise the manuscript at their discretion. Either way, the manuscript should not, in our opinion undergo further scientific review.

*Reviewer #1:*

This paper follows on the authors discovery of a role for the alternative mRNA cap-binding protein, 4EHP in effecting translational repression of specific mRNAs instructed by miRNA binding (at their 3' UTR). Here, the authors have used cells lacking this effector mechanism (4EHP knockout cells) to profile, in an unbiased manner, the consequences on translational regulation (separately from effects on mRNA abundance) and identified mRNAs whose translation is selectively regulated by miRNA in an 4EHP-dependent manner. This unbiased approach led them to the mRNA encoding an ERK phosphatase (*Dusp6*), which is normally repressed by miR-145 in a 4EHP-dependent manner. The strength of the paper derives from the unbiased approach to the discovery of mRNA targets of the 4EHP-dependent translational suppression mechanism and from the rigor by which the crucial mechanistic points have been made.

In addition to the data obtained by ribosome footprinting, de-repression of *Dusp6* in 4EHP∆ cells is shown to proceed via enhanced translation initiation, as the mRNA is observed to shift to heavier polysome fractions in the mutant cells.

The repression mechanism implicated not only 4EHP, but also is miRISC partners, CCR4, NOT and 4E-T (Figure 3). This is very convincing as evidence that 4EHP exerts its effect by the stated mechanism. The point is furthermore supported by the finding that 4EHP inactivation neuters the dominant effect of a miR-145 inhibitor (Figure 3). If there were a need to furnish further support for this point, one might consider comparing the ability of wildtype and mutant 4EHP to rescue the phenotype – provided mutations that affect the interaction between 4EHP and miRISC are known to exist.

The role of *Dusp6* de-repression in the phenotypic consequences of 4EHP inactivation is likewise well supported, both pharmacologically – with a DUSP6 enzymatic inhibitor and genetically.

All in all this is a study that can, in my opinion, be published as is.

*Reviewer #2:*

Manuscript "Translational control of ERK signaling through miRNA/4EHP-directed silencing" by Jafarnejad et al., shows an interesting story of how 4EHP as a component of miRISC controls DUSP6 expression and more importantly what is biological consequence of this control – ERK1/2 phosphorylation. The manuscript is clearly written and experiments support finding of the authors so I recommend publication of the manuscript.

*Reviewer #3:*

The manuscript describes research on a role of the eIFE-Homology Protein (4EHP) in translational regulation of cellular mRNAs. 4EHP was recently reported to act as a downstream effector of miRNA repression but the conclusions of that work were exclusively based on the analysis of artificial reporters responding to miRNAs. In the submitted work, the authors now carry out ribosome footprint profiling of cellular mRNAs using 4EHP knockout and control cells, and a number of additional biochemical and cell growth assays. The three most important conclusions of this work are: (I) demonstration that 4EHP indeed regulates expression of many endogenous mRNAs, among them dozens of mRNA with enrichment of miRNA site of the 3'-UTRs; (II) expression of many identified mRNAs is regulated at the level of translation with no effect on mRNA decay. The repression is associated with increased 4EHP interaction with the mRNA cap at the cost of diminished association of eIF4E; (III) for one of the targets, the DUSP6 phosphatase mRNA, the authors demonstrates that its repression indeed occurs via the miRNA-CCR4/NOT-4EHP axis and they identify miR-145 as a repressive miR, which targets a defined region in the DUSP6 3'UTR; (IV) It is demonstrated that miR-145 repression of DUSP6 results in increased phosphorylation of ERK1/2 which in turn results in increased cell growth and apoptosis.

These are all important findings, and in the view of many controversies regarding the mechanism of miRNA-mediated repression, the convincing evidence that endogenous miR targets are physiologically repressed via the CCR4/NOT-4EHP axis and that repression occurs the level of translation without the effect on mRNA stability, make the data particularly interesting. Clearly, the data validate the miRNA-CCR4/NOT-4EHP axis as the repression mechanism operating in mammalian cells and validate importance of the miR-145-mediated repression of DUSP6 in increased phosphorylation of ERK1/2 and its downstream effects.